# Understanding the light induced hydrophilicity of metal-oxide thin films

Rucha Anil Deshpande [1,3], Jesper Navne [1,3], Mathias Vadmand Adelmark [1], Evgeniy Shkondin [1], Andrea Crovetto [1], Ole Hansen [1], Julien Bachmann [1,2] & Rafael Taboryski [1] ✉

Photocatalytic effects resulting in water splitting, reduction of carbon dioxide to fuels using solar energy, decomposition of organic compounds, and light-induced hydrophilicity observed on surfaces of various metal oxides (MOx), all rely on the same basic physical mechanisms, and have attracted considerable interest over the past decades. TiO$_2$ and ZnO, two natively n-type doped wide bandgap semiconductors exhibit the effects mentioned above. In this study we propose a model for the photo-induced hydrophilicity in MOx films, and we test the model for TiO$_2$/Si and ZnO/Si heterojunctions. Experimentally, we employ a wet exposure technique whereby the MOx surface is exposed to UV light while a water droplet is sitting on the surface, which allows for a continuous recording of contact angles during illumination. The proposed model and the experimental techniques allow a determination of minority carrier diffusion lengths by contact angle measurements and suggest design rules for materials exhibiting photocatalytic hydrophilicity. We expect that this methodology can be extended to improve our physical understanding of other photocatalytic surface effects.

The transition metal oxides (MOx), titanium dioxide (TiO$_2$), and zinc oxide (ZnO), belong to a class of wide-bandgap natively doped n-type semiconductors[1,2] with several remarkable and widely studied properties[3–6]. The reported applications of TiO$_2$ and ZnO surfaces date back several decades and comprise solar energy conversion by single-crystal electrodes for water splitting[7–9], photocatalytic reduction of carbon dioxide to fuels using solar energy[10], photocatalytic decomposition of organic compounds[2,11,12], photo-electrodes for dye-synthesized solar cells[13,14], and photo-induced hydrophilicity[3,6,15–18]. The common root cause for all those effects of TiO$_2$, ZnO, and other MOx surfaces is the activation of chemically inactive surface groups when the materials are illuminated with ultraviolet (UV) light[3]. The activation happens through the excitation of electron−hole pairs that diffuse to the surface and engage in surface reactions, where, e.g., the holes can cause oxidation of adsorbed water molecules through the reaction $H_2O + h^+ \rightarrow \cdot HO + H^+$, whereas electrons can cause

reduction of oxygen by the reaction $O_2 + e^- \rightarrow \cdot O_2^-$. The formed radicals may then engage in further reactions at the surface[2].

Which surface groups get activated in MOx thin films by UV light is debated, and most likely, more than one unitary reaction is involved[4,19,20]. One possibility for anatase TiO$_2$ is the activation of Ti−O−Ti bridges. When a water molecule chemisorbs across one Ti−O bond, then one obtains two terminal Ti−OH groups[21], and those could be an example of what we call active surface groups. However, no comprehensive physical model exists for the photocatalytic wetting phenomena, and most of the reported experimental studies of the aforementioned effects have been carried out with lamps, e.g., mercury lamps and fluorescent light bulbs, generating spectra comprising multiple lines in the UV range, and without considering the effects of light propagation across both MOx thin films and substrates[22–26]. Today, the availability of monochromatic and bright UV light emitting diodes (UV LEDs)[27], and ultra-precise atomic layer deposition (ALD) of

[1]Technical University of Denmark, DTU Nanolab, National Centre for Nano Fabrication and Characterization, Ørsteds Plads B347, DK-2800 Kgs Lyngby, Denmark. [2]Friedrich-Alexander-Universität Erlangen-Nürnberg, Chemistry of Thin Film Materials, IZNF, Cauerstr. 3, 91058 Erlangen, Germany. [3]These authors contributed equally: Rucha Anil Deshpande, Jesper Navne. ✉e-mail: rata@dtu.dk

sub-wavelength thin films of MOx[28], allow us to develop and test a theoretical model for the phenomenon and carry out a more holistic and thorough study of the UV light-induced hydrophilicity, in which MOx/Si heterostructures are illuminated with collimated UV light with a narrow spectral linewidth and well-defined intensity.

The two materials TiO₂ and ZnO have many properties in common, however, ZnO is a direct-bandgap semiconductor[1,29] while TiO₂ in the (most photocatalytically relevant)[19,21,30] anatase structure has an indirect bandgap[31]. Moreover, whereas for TiO₂, native point defects such as oxygen vacancies and Ti interstitials account for the n-type conductivity of the materials with donor levels of order $10^{18}$ cm$^{-2}$ [32], the origin of the native n-type conductivity in ZnO is more likely due to the incorporation of hydrogen atoms as shallow donors[33,34], as the point defect levels for this material are located too deep in the bandgap for thermal excitation at room temperature[35,36]. Both materials can readily be deposited on fused silica and silicon substrates by various methods. Both TiO₂/Si and ZnO/Si heterojunctions are staggered (type II) heterojunctions, where the smaller bandgap Si has its conduction band edge above the conduction band edges of both TiO₂ and ZnO (Fig. 1a)[3,37]. The similarities and differences allow us to compare the two materials using the same fabrication and characterization methods (Fig. 1b,c). As seen in Fig. 1b, the fabrication by ALD allows for thin films with high uniformity across whole wafers of diameter 100 mm and 150 mm (see also Supplementary Note, Supplementary Fig. 1 for data on surface roughness, and Supplementary Fig. 2 for data on crystallinity). The UV light-induced wettability is reversible (Fig. 1d), and it is an excellent and readily measurable (Fig. 1c) proxy for all the surface effects mentioned above. However, as we shall see, the photoinduced wettability depends on the optical constants in both the MOx and the Si substrate. Further, our findings indicate that the

photoinduced wetting is most pronounced for specific MOx layer thickness $d$ that optimizes the absorption due to internal interference effects in the MOx layers. These internal reflections cause an oscillatory dependence of the absorptance (absorbed optical power to incident optical power) on MOx layer thickness $d$, while also electron–hole pair diffusion lengths and heterojunction band-bending effects play important roles.

Our study demonstrates how MOx heterostructures can be precisely engineered in terms of layer thickness to yield optimal photocatalytic performance at specific illumination wavelengths namely 365, 300, 275 nm spanning a spectral range from near UV to deep UV. To show its practical application for open-surface microfluidic channels, we show in Fig. 1e a formation of a light-induced microchannel by dipping in water a sample that was illuminated by UV light through a photomask[38]. The study can be extended to other material systems with similar photocatalytic properties if the optical constants, band alignments, and layer thicknesses are known.

## Theoretical model for photocatalytic wettability in metal oxide/Si heterojunctions

Our starting point is the Cassie–Baxter model for the wettability of chemically heterogeneous surfaces, which yields the contact angle between a liquid and a solid surface that minimizes the Gibbs free energy for the triple phase system composed of the solid surface, the liquid, and the surrounding gas[39]. Gibbs free energy depends linearly on the wetted area[40], and hence, for a planar surface comprising two distinct surface chemistries, such that a surface fraction $f$ has surface chemistry corresponding to contact angle $\theta_0$ (say the intrinsic contact angle before UV illumination), whereas the remaining fraction $1 - f$ has surface chemistry corresponding to another contact angle $\theta_1$ (say the

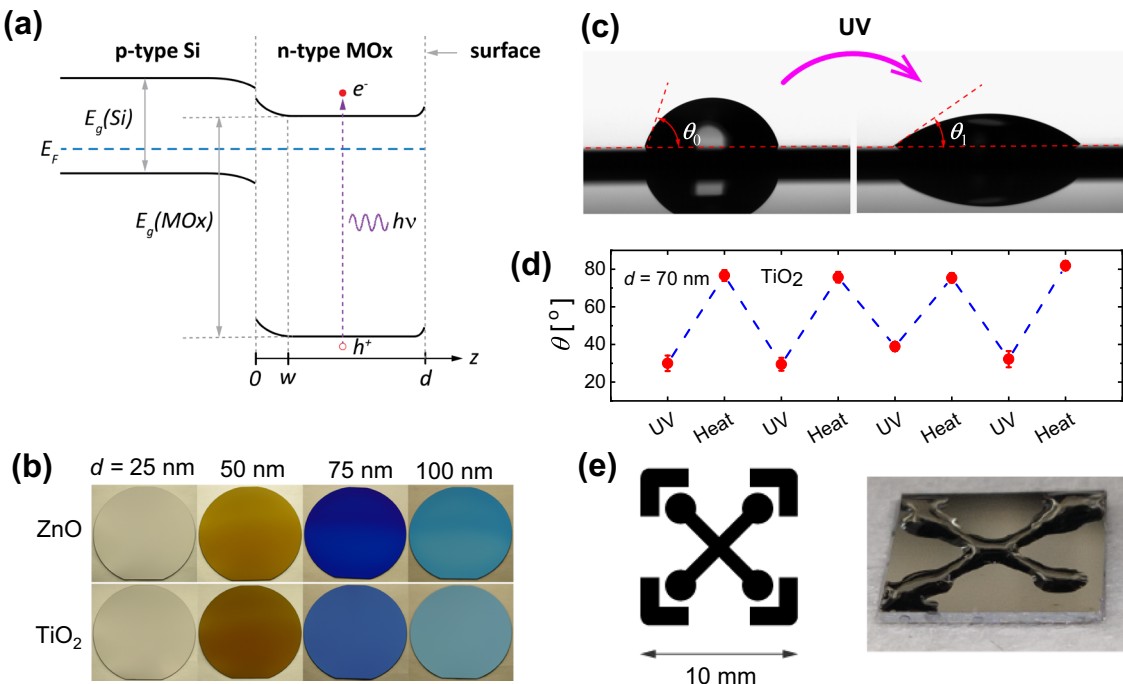

**Fig. 1 | Properties and features of the studied ZnO/Si and TiO₂/Si heterojunctions. a** Schematic band diagram of a type II MOx/Si heterojunction. Boron-doped, p-type Si is used as the substrate for ALD deposition of metal oxides, ZnO and TiO₂, which are natively n-type doped. The alignment of the Fermi levels creates the band banding in the MOx. The heterojunction band alignment is staggered (type II). **b** ALD allows for the deposition of precise MOx layer thicknesses, here illustrated by the colors of 100 mm Si wafers with different MOx thicknesses $d$ as indicated. **c** Demonstration of the hydrophilic effect for MOx induced by UV light at 365, 300,

and 275 nm wavelengths. The image shown is for water droplets before (left), and after (right) UV illumination of a TiO₂ surface with 365 nm light. **d** Restoration after UV exposure (365 nm @3.2(4) J cm$^{-2}$) can be done by annealing for one hour at ~100 °C. The advancing contact angle is shown here for a 70 nm thick TiO₂ layer on a Si substrate. Error bars represent standard deviations (STD) with a minimum of three sample measurements from the same wafer. **e** Illumination of the MOx/Si heterojunctions by UV light through a mask (left) allows for selective wetting of complex geometries (right) upon dipping in water.

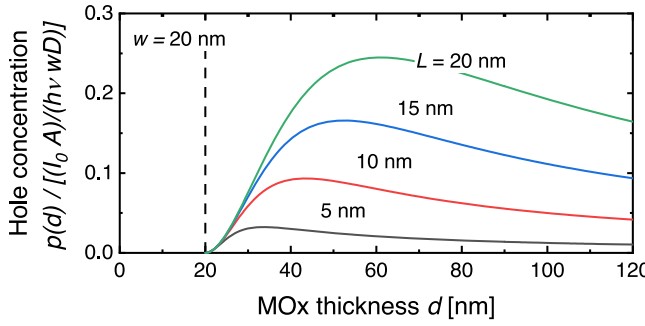

**Fig. 2 | Computed concentration of photo generated holes at the MOx surface.** The concentration of photo generated holes at the surface is plotted as a function of MOx layer thickness in the limit of low surface recombination with $s = 0$ computed with Eq. 6 for relevant values of depletion layer width $w$ and hole diffusion length $L$. $p(d)$ is expressed in units $(I_0 A)/(h\nu Dw)$, i.e., what is plotted in the figure is the dimensionless function $L^2/wd(1 - 1/\cosh((d - w)/L))$, with $w = 20$ nm.

saturation contact angle obtained after UV illumination for a long time), the Cassie-Baxter equation[39] predicts a minimum in interface energy for a contact angle given by:

$$\cos\theta = f \cos\theta_0 + (1 - f)\cos\theta_1. \tag{1}$$

As explained above, when illuminating the MOx surface by UV light, we excite electron-hole pairs in the MOx layer that diffuse to the surface and gradually activate surface groups corresponding to the contact angle $\theta_1$. In the beginning, the generated electron-hole pairs have plenty of surface groups to activate from $\theta_0$ to $\theta_1$, however, by continuing the illumination, eventually, all surface groups will become activated.

Several competing chemical reactions with separate rate constants may take place when the MOx is illuminated with UV light[5,41]. However, here we propose a simplified model assuming that the reaction kinetics for the chemical reactions at the surface are first order in both the area concentration of inactivated surface groups $N_s$ (units of m$^{-2}$), and the photo-generated excess carrier concentration at the surface $p(z = d)$ (units of m$^{-3}$). We then get:

$$\frac{\partial N_s}{\partial t} = - kp(d)N_s \simeq -N_s/\tau, \tag{2}$$

where $k$ (units of m$^3$ s$^{-1}$) is the rate constant. The single time-constant approximation is valid if the switching rate does not significantly affect (deplete) the surface carrier concentration. Then the time-constant is $\tau = (kp(d))^{-1}$. For an n-type MOx semiconductor, holes are the minority carriers, electrons are plentiful, and for a wide bandgap semiconductor, it is safe to assume that the light-induced excess carrier concentration is approximately equal to the total hole concentration under illumination conditions. Since electrons and holes are photo-generated in pairs, the photoinduced electron concentration will be of order, $\sim (I_0\tau_r)/h\nu d$. The recombination time is typically not known, but for $I_0 \sim 100$ mW cm$^{-2}$, $d \sim 50$ nm, a photon energy $h\nu \sim 3$ eV, and assuming a relatively long recombination time of order 1 μs, we get $p \sim 5 \times 10^{16}$ cm$^3$, which is two orders of magnitude lower than the native doping level in TiO$_2$. Given that the recombination is, in fact, likely faster than 1 μs, $p$ is likely to be much lower, and the assumption introduces no significant error. Hence $p(z)$ will be referred to as simply the hole concentration. If the switching process is the only surface recombination (loss) mechanism for holes at the surface, and one hole is consumed per activated surface group, we have $\frac{\partial N_s}{\partial t} = - sp(d) \approx - N_s/\tau$ where $s$ is the surface recombination velocity of holes at the MOx surface. If the activation rate is low, it follows that the surface velocity

is also low, and then the single time-constant approximation is valid. Equation 2 is readily solved by:

$$N_s(t) = N_s(0)\exp(-t/\tau), \tag{3}$$

where $N_s(0)$ is the initial area concentration of inactivated surface groups. Identifying $f(t) = N_s(t)/N_s(0)$ in Eq. 1, where $t$ is the illumination time, we get:

$$\cos\theta(t) = e^{-t/\tau}\cos\theta_0 + \left(1 - e^{-t/\tau}\right)\cos\theta_1. \tag{4}$$

To derive the hole concentration at the surface $p(d)$, we choose $z = 0$ at the Si/MOx interface and let $z = d$ (see Fig. 1a) at the MOx surface. Assuming flat-band conditions, the hole concentration $p(z)$ in the illuminated semiconductor obeys the continuity equation:

$$D\frac{\partial^2 p(z)}{\partial z^2} - \frac{p(z)}{\tau_r} + g(z) = 0, \tag{5}$$

where $g$ is the illumination induced generation rate, $D$ is the diffusivity, and $\tau_r$ is the excess carrier recombination lifetime in the MOx; for future, we also define the diffusion length $L = \sqrt{D\tau_r}$. We now introduce the boundary conditions for Eq. 5. The contribution from holes generated at $z < w$ (where $w$ is the MOx/Si depletion region width) is assumed to be zero and the boundary condition at $z = w$ must reflect the fact that holes in the MOx in proximity of the depletion layer have a very high probability to transfer to the silicon, which leads to a very high recombination velocity at the depletion layer edge (probably of order of $\sim 10^7$ cm s$^{-1}$), and thus a fair approximation is to take the limit of infinitely fast recombination velocity corresponding to $p(w) = 0$. At the surface we have $-D\frac{\partial p}{\partial z}\Big|_{z=d} = sp(d)$. For simplicity, we consider only absorption in the MOx, and a uniform generation rate $g(z) = g_0 = I_0 A/(dh\nu)$, where $I_0$ (units of W m$^{-2}$) is the illumination intensity, $A$ the absorptance (dimensionless), and $h\nu$ the photon energy. We expect the uniform generation assumption to be reasonable if $d$ is not too large compared to $1/\alpha$, where $\alpha$ is the absorption coefficient of the MOx. We then get the hole-concentration at the surface by solving Eq. 5 with the stated boundary conditions and inserting $z = d$.

$$p(d) = g_0\tau_r \frac{\cosh(\frac{d-w}{L}) - 1}{\cosh(\frac{d-w}{L}) + \frac{sL}{D}\sinh(\frac{d-w}{L})} \xrightarrow{s \to 0} g_0\tau_r\left(1 - \frac{1}{\cosh(\frac{d-w}{L})}\right). \tag{6}$$

In Fig. 2, we plot $p(d)$ from Eq. 6 for relevant values of $w$ and $L$ in units of $(I_0 A\tau_r/wh\nu)$ and in the low surface recombination limit at the MOx/air surface, setting $s = 0$.

The switching rate may now be expressed as:

$$\frac{1}{\tau} = kp(d) = k\frac{I_0 A\tau_r}{h\nu d}\left(1 - \frac{1}{\cosh(\frac{d-w}{L})}\right) = \frac{I_0 A}{h\nu}\frac{k}{D}\frac{L^2}{d}\left(1 - \frac{1}{\cosh(\frac{d-w}{L})}\right), \tag{7}$$

where the constant $k/D$ has the dimension of length but no physical meaning per se, whereas the prefactor $I_0 A/h\nu$ expresses the number of absorbed photons per unit time and per unit area in the MOx.

We now derive the dependence of the absorptance on the MOx layer thickness, $A(d)$. Although, the electron-hole pairs diffuse with positive direction towards the surface, which is why it was natural to put $z = 0$ at the MOx/Si interface in the derivation of Eq. 7, light propagation across the heterojunction starts with incidence on the surface, and hence, it is more natural to choose $z = 0$ at the surface, and positive direction towards the MOx/Si interface. Due to differences in Fresnel coefficients, light waves with incidence on the MOx surface, will undergo reflections at both the MOx/air interface and at the MOx/

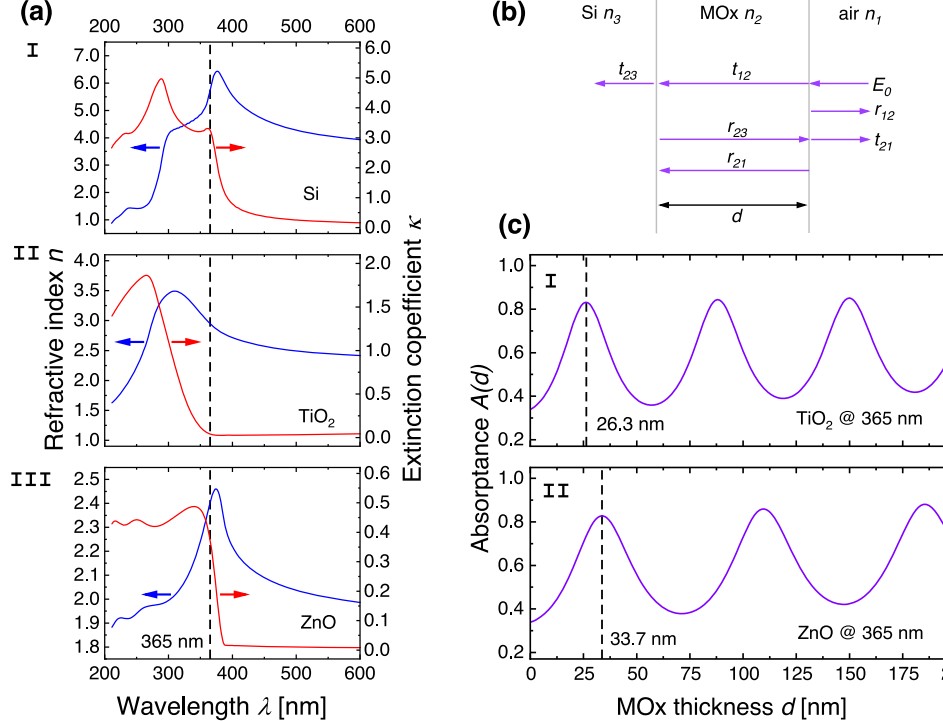

**Fig. 3 | Computation of the light absorptance due to multiple internal reflections in the MOx layer based on measured values of the refractive index and the extinction coefficient. a** Measured optical constants, refractive indices $n$ (blue curves), and extinction coefficients $\kappa$ (red curves) as functions of wavelength for I Si, II TiO$_2$ (anatase), and III ZnO for computation of the Fresnel coefficients in Eq. 8. The dashed vertical line indicates the position of the 365 nm wavelength. **b** Schematic illustration of the reflection $r$ and transmission coefficients $t$ from Eq. 8 for an incident wave with electric field amplitude $E_0$. The indices $j$ of the optical

constants $\widetilde{n}_j$ for the three media refer to air (index 1), MOx with thickness $d$ (index 2), and the Si substrate (index 3). **c** The absorptance as a function of the MOx layer thickness $d$ for illumination with UV light at 365 nm wavelength for TiO$_2$/Si (I) and ZnO/Si (II). The curves are computed with Eq. 9. The vertical dashed lines indicate the positions in MOx thickness yielding the first maxima in absorptance for the two heterojunctions. (See also Supplementary Fig. 3 for computed absorptance at other wavelengths).

Si interface. Moreover, the internal reflections will result in a Fabry–Pérot type of interference effect[42] that yield an oscillatory behavior of the reflected, transmitted, and absorbed light intensity with MOx layer thickness. The Fresnel coefficients for normal incidence that yield the transmission coefficients $t_{jk}$ from medium $j$ to medium $k$, and reflection coefficients $r_{jk}$ in medium $j$ at the interface to medium $k$, are given by:

$$t_{jk} = \frac{2\widetilde{n}_j}{\widetilde{n}_j + \widetilde{n}_k} \wedge r_{jk} = \frac{\widetilde{n}_j - \widetilde{n}_k}{\widetilde{n}_j + \widetilde{n}_k}, \tag{8}$$

with the complex generalized indices $\widetilde{n}_j = n_j + i\kappa_j$, where $n_j$, the refractive index and $\kappa_j$, the extinction coefficient, are thickness independent material parameters. In Fig. 3a, we have plotted measured values of the refractive indices and extinction coefficients for the three materials, silicon (I), anatase TiO$_2$ (II), and ZnO (III). In the following, we use the notation $j = 1$ for air, $j = 2$ for MOx, and $j = 3$ for Si (see Fig. 3b). From the Fresnel coefficients in Eq. 8, we can compute the total reflectance $R(d)$ and total transmittance $T(d)$. After some algebra we obtain Eq. 9 (see Supplementary Derivation).

$$R(d) = \left| r_{12} + \frac{t_{12}r_{23}t_d^2 t_{21}}{1 - r_{23}t_d^2 r_{21}} \right|^2 \wedge T(d) = \left| \frac{t_{12}t_{23}t_d}{1 - r_{23}t_d^2 r_{21}} \right|^2, \tag{9}$$

where $t_d = e^{\frac{i2\pi \widetilde{n}_2 d}{\lambda}}$ is the phase factor picked up from traversal of the MOx layer. The absorptance, i.e., the absorbed intensity to incident intensity is then obtained as $A(d) = 1 - R(d) - T(d)$. In Fig. 3c we have plotted the absorptance as a function of MOx layer thickness $d$ for the

two heterojunctions Si/TiO$_2$ (I), and Si/ZnO (II) for illumination with 365 nm wavelength UV light. For a comparison of absorptance at all switching wavelengths, 365, 300, and 275 nm, in this study, see Supplementary Fig. 3. In the plots, we also indicate the MOx thicknesses corresponding to the first absorptance maximum. Due to the decay with $1/d$ of the hole concentration $p(d)$ (Fig. 2), we only expect the first couple of maxima to play a role for the switching rate computed with Eq. 7.

## Results and discussion

### Contact angle switching rate

After verifying that the TiO$_2$ and ZnO films are in the expected anatase and wurtzite phases (Supplementary Fig. 2), we test the proposed model for photocatalytic wettability. We consider Eq. 4 and Eq. 7, where the direct contact angles and their rate of change can be measured. We do this by placing a water droplet, exposing it to UV light from a collimated UV LED source, and simultaneously recording its contact angle switching, as shown schematically in Fig. 4a. Traditionally, such experiments are carried out by exposing the samples to different UV illumination doses in the dry state using mercury lamps in bulky chambers and a subsequently measuring of their contact angles in a goniometer setup. In our study, we distinguish these methods by calling them "wet" and "dry" exposures. The availability of compact light sources such as UV LEDs allows for the practical wet exposure measurements of the photo activity, as shown in Fig. 4a. Both the dry and wet exposure experimental setups with detailed images are shown in Supplementary Fig. 4. Examples of such measurements are shown in Fig. 4b for (I) ZnO/Si and (II) TiO$_2$/Si. Due to pinning effects, which are also responsible for the contact angle hysteresis[43,44], the decrease of

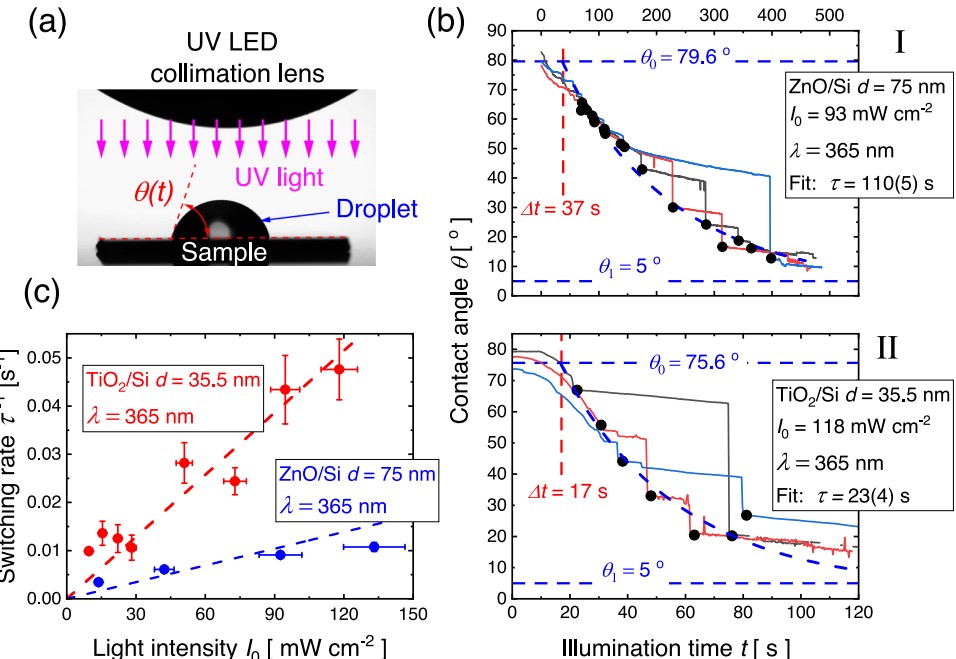

**Fig. 4 | Obtaining the contact angle switching times by the wet exposure method. a** Profile image of the droplet, sample, and LED collimation lens (above) from the contact angle goniometer with self-explanatory graphics added to show the key elements in the wet exposure experiment. **b** Example of apparent contact angles measured during wet exposure by UV light with a wavelength of 365 nm (for illumination with 300 and 275 nm UV light, see Supplementary Fig. 5). The apparent contact angles switch to lower values in steps. The contact angle values right after the switches are marked with black dots and used for fitting to Eq. 4. Three traces for each MOx layer thickness are used to generate a fit of one switching time $\tau$. In (I) the fitting is shown for a ZnO/Ti heterojunction with a ZnO layer thickness of 75 nm, whereas in (II) the fit is shown for a TiO$_2$/Si heterojunction with a TiO$_2$ layer of

35.5 nm. The onset of photoinduced switching happens with a delay time $\triangle t$, where a smooth decay is observed, which is associated with photocatalytic oxidation of adsorbed hydrocarbon groups on the surface. $\triangle t$ is fitted alongside the switching time constant and the limiting contact angles $\theta_0$ and $\theta_1$. **c** Validation of linearity of switching rate $1/\tau$ with light intensity $I_0$. Shown data are for the same samples as in (**b**) and also illuminated at 365 nm. Error bars in switching rate represent one STD to the fit of switching time constant as shown in (**b**) with a minimum of three samples, while error bars in light intensity represent accumulated errors from intensity variation during the illumination and uncertainty in the determination of illumination area (see Methods).

the contact angle during illumination takes place stepwise. The abrupt switches to smaller contact angles are stochastic processes and occur when the surface/droplet system matches the maximum sustainable contact angles, i.e., corresponding to the advancing contact angle values[45]. The slow linear decrease in the steps is due to the evaporation of the droplet. The data could only be reliably fitted to Eq. 4 if a delay time $\Delta t$ was introduced for the onset of photocatalytic/induced switching. We believe this delay time may partly be associated with removal by photocatalytic oxidation of hydrocarbon surface contaminations adsorbed on the surfaces during storage at ambient conditions[46,47], but also experimental error where a time delay of a few seconds is expected when manually placing the water drop and manually turning on lamp and camera.

Within experimental uncertainty, the rate $1/\tau$ of contact angle switching increased linearly with illumination intensity, as predicted by Eq. 7. Example of this is shown for two of the samples in Fig. 4c. Deviations from this linear dependence may be expected in the case of saturation, where close to all hydrophilic surface groups become activated. Some saturation in the switching rate as a function of light intensity may perhaps be spotted for ZnO/Si in Fig. 4c.

Equation 7, which is the expression for the switching rate $(1/\tau)$, further includes important parameters for photocatalytic wetting, namely the surface hole-concentration $p(d)$, and the absorptance $A(d)$ of the UV light. To characterize these parameters, we fitted the activation times for different thicknesses and wavelengths as shown in Supplementary Fig. 5. While the switching rate is likely influenced by the film's microstructure, we have verified (Supplementary Fig. 2) that the main microstructural features of the films are roughly thickness independent. Hence, Eq. 7 can be applied to different film thicknesses.

To simplify the comparison with Eq. 7, we divided the fitted switching rates by the individual light intensities, $I_0$ used in the experiments. This was done for both TiO$_2$/Si and ZnO/Si heterojunctions at three wavelengths, 365, 300, and 275 nm and the analyzed results are shown in Fig. 5. The output power (24 mW) of the 300 nm LED was, however, not sufficient to trigger switching of the contact angle in ZnO/Si within approximately one hour of illumination, which is why no data is plotted for the ZnO/Si heterojunctions with illumination at 300 nm. The comparison between TiO$_2$/Si and ZnO/Si heterojunctions reveals that the surface site activation rate is about a factor of 20 higher for the TiO$_2$/Si heterojunctions than for the ZnO/Si heterojunctions. Moreover, the dependences on illumination wavelength are in stark contrast. While the TiO$_2$/Si heterojunctions exhibit the fastest contact angle switching (highest activation rate) for the shortest wavelengths, 275 nm and 300 nm, the opposite is the case for the ZnO/Si heterojunction, which exhibits the most efficient switching for illumination at 365 nm. This may indicate that different types of surface groups get activated upon illumination with UV light for the two heterojunctions or differences in surface collection efficiencies for the two materials. Regarding the depletion layer width $w$, we fitted all data in Fig. 5 with the same $w$ for all layer thicknesses for simplicity, although samples with different thicknesses were fabricated on different wafers. Although the wafers came from the same batch, differences in doping may occur as the batch had a doping specification in terms of conductivity of 1-20 $\Omega$cm, which corresponds to a span in doping levels of more than one order of magnitude[48]. However, there is some indication of the depletion width being consistently higher for the ZnO/Si heterojunctions than for their TiO$_2$/Si counterparts, which might indicate a higher level of native n-type doping in TiO$_2$ over ZnO in this

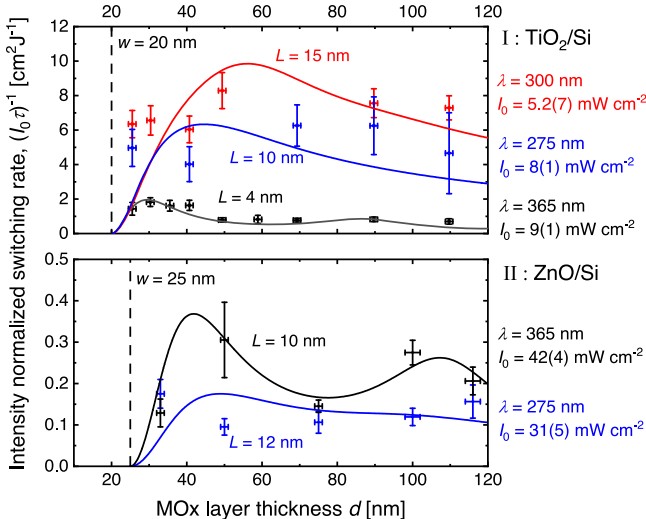

**Fig. 5 | The contact angle switching rates for TiO₂/Si and ZnO/Si heterojunctions as a function of the MOx layer thicknesses.** Switching rates ($\tau^{-1}$) resulting from fits to Eq. 7 and normalized with the respective illumination intensities ($I_0$) plotted vs. the MOx layer thicknesses for the two heterojunctions, TiO₂/Si (I), and ZnO/Si (II) are shown as datapoints with error bars. The error bars represent the standard deviation from three samples with a nominal but experimentally verified layer thickness. For both types of junctions, results from illumination by UV light at 365, 300, and 275 nm are shown. For ZnO, the 300 nm LED had too low intensity to trigger switching of the contact angle within a reasonable time. The solid curves represent fits to Eq. 7, after division by $I_0$ on both sides, where $w$, $L$, and $k/D$ were used as adjustable parameters, and where for each of the heterojunctions, the depletion layer widths $w$ were set to the same value, whereas the diffusion length $L$, and quantity $k/D$, were allowed to vary, both with material and illumination wavelength. $k/D$ values adjusted to fit the data were the following: $k/D$(TiO₂/Si, 300 nm) = $3.0 \times 10^{-13}$ m, $k/D$(TiO₂/Si, 275 nm) = $4.0 \times 10^{-13}$ m, $k/D$(TiO₂/Si, 375 nm) = $3.0 \times 10^{-13}$ m, $k/D$(ZnO/Si, 375 nm) = $1.8 \times 10^{-14}$ m, and $k/D$(TiO₂/Si, 275 nm) = $7.0 \times 10^{-15}$ m. Due to the number of fitting parameters and the uncertainties of the $(I_0\tau)^{-1}$ data, the uncertainty of the fits yielding $w$, $L$, and $k/D$ are estimated to be in the order of ±25%. Vertical error bars represent one STD from three samples, whereas error bars in $d$ represent measured layer thickness variation (maximum−minimum) from five locations across each wafer.

case. The fitting of $w$ is, however prone to uncertainties on the order 25%, and the depletion width also depends on the interface native oxide, which was not removed before the deposition of MOx by ALD, which is why firm conclusions cannot be drawn.

The obtained hole-diffusion lengths are of order $L \sim$ 5–15 nm for both materials. It is challenging to compare those values with values for diffusion lengths reported in the scientific literature, as everything from excition diffusion lengths in polythiophene-sensitized TiO₂ bilayers of below 6 nm[49], over $\sim$ 100–500 nm for minority carriers in bulk ZnO materials[50–52], to electron diffusion lengths in TiO₂ of order 10–30 μm have been reported[53,54]. However, given the polycrystalline nature of the thin MOx layers, their high hole effective masses, and room temperature measurement conditions, a short mean free path for carrier transport is expected, and we believe the found values for $L$ in this work are realistic. A possible explanation for the low $L$ at 365 nm is that this wavelength is not sufficient to excite carriers across the nominal band gap but only into low-mobility tail states.

## Comparison of dry and wet exposure
Next, we discuss the difference between dry exposure measurements and wet exposure measurements for the two materials. In the wet exposure mode, a water droplet was placed on the surfaces during illumination, whereas in the dry exposure mode, the surfaces were illuminated with a given dose of light prior to determining the

advancing contact angle. The wet exposure mode is expected to favor the chemical pathway, where holes cause oxidation of water molecules through the reaction $H_2O + h^+ \rightarrow \cdot HO + H^+$, whereas the dry exposure mode is expected to favor the chemical pathway, where electrons cause reduction of adsorbed oxygen by the reaction $O_2 + e^- \rightarrow \cdot O_2^-$. This is simply because the surface is exposed to water under wet conditions, and to air ($O_2$) under dry conditions. Although both pathways clearly are active, our study indicates that the hole-induced reaction is strongest. This is supported by the observation that wet exposure experiments always showed shorter activation times compared to dry exposure experiments. An example of this is shown in Fig. 6, where the activation times for some of the fastest switching samples are shown, (I) TiO₂/Si with $d$ = 30 nm illuminated at 275 nm, and (II) ZnO/Si with $d$ = 100 nm illuminated at 365 nm.

## Significance of bandgaps and illumination wavelenghts
Our experimental data confirm the well-established origin of light-induced hydrophilicity, namely the excitation of electron–hole pairs by light with photon energies exceeding the bandgap of the metal oxides. We determined both the optical and electronic bandgaps for the two materials (see Fig. 7 and Supplementary Fig. 6). This analysis revealed that, except for very thin MOx layers below $w$ where the bandgap is prone to depend on quantum confinement effects, no bandgap dependence on MOx thickness $d$, within experimental uncertainty, was observed (Fig. 7b). We measured both the optical and electronic bandgaps values for the two materials ($E_g^{el}$(TiO₂) = 3.469(2) eV, $E_g^{opt}$(TiO₂) = 3.43(2) eV, $E_g^{el}$(ZnO) = 3.159(5) eV, $E_g^{opt}$(ZnO) = 3.27(4) eV). While the obtained values for ZnO are in line with reference values in the range 3.2-3.3 eV[55], the values for TiO₂ are somewhat higher than common reference values[3] of 3.2 eV. However, often higher bandgaps are encountered for TiO₂ and ZnO thin films when compared to their bulk values[37,56–58]. In Fig. 7c, we compare the measured bandgap energies for the two materials with the photon energies for the LEDs used for excitation. A reference experiment illuminating with 405 nm light showed no photoinduced switching of contact angles. This perfectly corroborates our expectations, as the photon energy for this LED is below the measured energy bandgaps for both materials. Interestingly, the 365 nm LED was able to induce switching of the contact angle in both materials. For ZnO, this is expected, as the photon energy of the 365 nm LED is comfortably higher than the measured bandgap of ZnO. However, for TiO₂, the measured optical energy bandgap is located on the tail of the photon energy of the 365 nm LED. The observed onset of photoswitching of the hydrophilicity for the TiO₂/Si heterojunction, albeit with low contact angle switching rates (see Fig. 5, TiO₂ data), at the 365 nm photon energy of 3.4 eV aligns well with the measured bandgap for this material (Fig. 7c).

## Implications of the results
The most important finding we report here is that photocatalytic surface phenomena of metal oxides strongly depend on the optical constants (the refractive index and extinction coefficient) of the materials and on the film thicknesses in a non-monotonic manner. In addition, our study reveals that the diffusion lengths of photo-generated carriers can be obtained from simple contact angle measurements. Those dependencies can be predicted within the framework of the presented physical model. Importantly, we can obtain design rules for the fastest possible switching surface. For example, using a wide bandgap insulator, such as SiO₂ as the substrate, should maximize the surface hole concentration $p(d)$ by mitigating recombination at the MOx/substrate interface. In addition, according to Fig. 2, $w$ should be minimized (so, higher doping density of the MOx or lower doping density of Si is beneficial), the bulk lifetime of the MOx should be maximized, and $D$ (i.e., the minority carrier mobility) should be minimized. The low-mobility design rule ensures that backside recombination with the substrate is minimized. This requirement is in

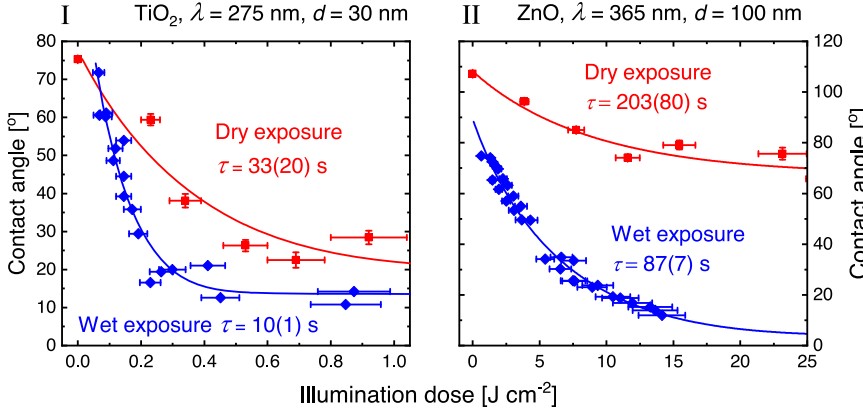

**Fig. 6 | Comparison of the contact angle switching rates obtained by dry exposure and wet exposure illuminations.** Comparison of dry exposure and wet exposure measurements for TiO$_2$/Si at $d = 30$ nm and $\lambda = 275$ nm (I) and ZnO/Si at $d = 100$ nm and $\lambda = 365$ nm (II). The fitted activation time values $\tau$ are shown in the figure. Vertical error bars for dry exposure represent one STD from a minimum of three samples, whereas horizontal error bars (both wet and dry exposure) represent accumulated uncertainties in dose determinations (see Methods).

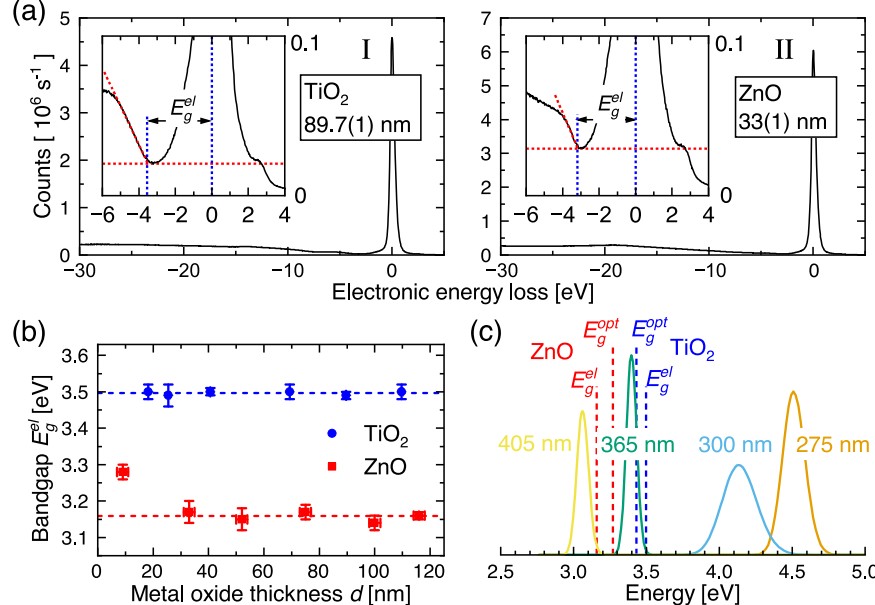

**Fig. 7 | Mapping of the measured TiO$_2$ and ZnO energy bandgaps to the photon energies of the LEDs used for activation of the hydrophilic surface states. a** The electronic bandgaps of the MOx layers were measured using reflection electron energy loss spectroscopy (REELS) for both TiO$_2$ (I) and ZnO (II) layers. The results are shown in (**b**) for both TiO$_2$ and ZnO for different layer thicknesses. The average bandgap over all thicknesses for TiO$_2$ was thus determined to be 3.469(2) eV, and 3.159(5) eV for ZnO, where the outlier at $d = 9$ nm was discarded. Vertical error bars represent accumulated errors from the band gap determination shown in (a), where the energy loss values from three samples were used. Horizontal error bars were obtained from measured layer thickness variation (maximum–minimum) from five locations across each wafer (see Methods). The measured electronic- and optical energy gaps for the TiO$_2$ and ZnO layers are compared to the normalized Gaussian power output distributions on photon energies for the UV LEDs used in the experiments (**c**). The Gaussian distributions are obtained from the center frequencies and full-width-half-maxima (FWHM) values in the LED specifications sheets from the supplier.

stark contrast to the typical requirement of high mobilities in photo-absorbing materials for photoelectrochemical and photovoltaic cells.

# Methods
## Substrates
The Si wafers used as substrates for the fabrication of the MOx/Si heterojunctions were prime quality 150 mm diameter, double-sided polished, ⟨100⟩ oriented, 500(20) μm thick, boron-doped (p-type) with a resistivity of 1–20 Ωcm, procured from Siegert Wafers. Wafers were used as received.

## MOx deposition
The TiO$_2$ and ZnO layers were deposited by atomic layer deposition (ALD) (Advanced Plasma ALD system, model R-200 with a plasma source manufactured by Picosun Oy). The deposition was done without plasma enhancement with a varying number of cycles to obtain different layer thicknesses. The reaction chamber was pumped down below 20 hPa during the process. For the deposition of TiO$_2$, the reactor chamber was stabilized at 300 °C to ensure anatase phase formation of TiO$_2$. Precursor 1 was TiCl$_4$, pulsed for 0.1 s and then purged for 4 s. Precursor 2 was H$_2$O, pulsed for 0.1 s and then purged for 5 s. Nitrogen gas with a flow rate of 150 sccm was used as the carrier

gas. For the deposition of ZnO, the reactor chamber was stabilized at 200 °C. Precursor 1 was diethylzinc (DEZ), pulsed for 0.1 s and purged for 5 s. Precursor 2 was $H_2O$ with similar pulse and purge times. A Nitrogen gas with a flow rate of 150 sccm was used as the carrier gas during precursor 1 and with a flow rate of 200 sccm during precursor 2.

## Use of photomask for patterned illumination

The patterned wetted area shown in Fig. 1e was obtained by illumination of the sample with a high dose at 365 nm UV light through a darkfield polarity photomask followed by dip coating of the sample in water. The photomask was designed using Clewin vs. 4 software and procured from Compugraphics (Glenrothes, UK).

## Determination of MOx layer thicknesses

Layer thickness verifications were done using optical ellipsometry to estimate their thickness values with a variable angle spectroscopic ellipsometer (VASE, model M2000XI-210, procured from J. A. Woollam Co.). The measurements were performed with collimated polarized light in the spectral range 210-1690 nm incident on the sample with a spot size of approximately 2 mm for different incident angles from 40° to 70°. Data were analyzed with a model composed of a similar stack of materials to extract the layer thickness corresponding to the least errors in the fit using the completeEASE software. The thickness variation across the wafer was assessed by measuring the thickness on multiple areas of the wafer. The thicknesses of all samples deposited using ALD varied in the range of 1–4 nm across the full wafer.

## Crystallinity of MOxs

Grazing-Incident X-ray Diffraction (GiXRD) analysis was performed with a Rigaku SmartLab diffractometer using Cu Kα radiation and a parallel beam geometry. The incident X-ray beam remained fixed at a low incident angle of 0.5°, while the detector was moved around the sample to conduct 2θ scans.

## Measurement of optical constants for the materials

The optical constants of ZnO, $TiO_2$, and the Si substrate were determined using the ellipsometry method (VASE, model M2000XI-210, procured from J. A. Woollam Co.). To extract the optical constants of $TiO_2$, the Tauc-Lorentz model was employed, with the inclusion of two additional Lorentz oscillators, which enabled a suitable fit within the wavelength range of 210 nm to 1690 nm.

For ZnO, a similar approach using the Tauc-Lorentz model was adopted. However, to achieve a better fit within the shortest wavelength range of 210 nm to approximately 400 nm, an additional Drude term was introduced.

Regarding the Si substrate, the optical constants were retrieved by analyzing the pseudo dielectric function measured directly on a wafer piece with a single-side polished surface devoid of native oxide. This pseudo dielectric function is related to the measured ($\Psi,\Delta$) values using the following equation:

$\varepsilon = \langle \varepsilon \rangle = \sin^2\theta_i(1 + \tan^2\theta_i(\frac{1-\rho}{1+\rho})^2)$, where $\rho = \tan\Psi e^{i\Delta}$ and $\theta_i$ is the angle of incidence[51]. It is worth noting that when the model represents a perfectly flat substrate with infinite thickness, the physical dielectric function is equal to the ellipsometry-measured pseudo dielectric function. The relation between the complex dielectric function and complex refractive index is given by the following relation: $\tilde{n}^2 = \varepsilon$. One way to verify if the pseudo dielectric function is indeed an accurate representation of the real physical values is by conducting measurements at different incident angles. Subsequently, it can be confirmed that the calculated dielectric function remains independent of those angles.

## Atomic force microscopy (AFM) for morphology characterization

The AFM was performed in tapping mode at ambient conditions (Dimension Icon-PT AFM system procured from Bruker AXS with TAP150 probes from NanoAndMore), using high aspect ratio Si tips (radius of curvature <7 nm). In the tapping mode, we used a driving frequency of 170 kHz, an amplitude of 272 mW, a set point of 155 mV, and a scan rate of 0.977 Hz. The scan images were acquired in 1 μm² area on a sample at two different regions to obtain reliable measurements. The images were processed in Nanoscope Analysis software and corrected for flatness. The roughness of the samples is expressed as the average $R_q$ values for all measurements on the sample, which equals the square root of the sum of squares of individual heights. The difference between the surface area of the 3D imaged profile and the projected area (the 2D image) of the sample is also extracted from the AFM data of the thin films. The AFM data is shown in Supplementary Fig. 1.

## Determination of MOx bandgaps

Reflection electron energy loss spectroscopy (REELS) was used to determine the band gaps (XPS Nexsa system, Thermofisher Scientific) at a pressure below $2 \times 10^{-7}$ mbar, and using the system's auto height feature. Surface impurities were removed by sputtering with a beam consisting of clusters of 150 Argon ions accelerated to 6 keV energy for 400 s with no significant etching of the material surface. REELS spectrums were acquired with energy steps of 0.03 eV with an initial beam energy of 1 keV and a pass energy of 10 eV. Counts were averaged over 4 scans and dwell time of 50 ms. A linear model adapted from Vos et al.[59]. was used to interpret the energy loss spectrum immediately after the dip assigning the interception of that line with the energy axis as the bandgap (see Fig. 7a). This calculation was performed with a Python script processing the measurements in the following steps: (1) Subtracting the elastic energy peaks from all energy measurements to obtain the electronic energy loss, (2) determining the background counts as the minimum value in the dip and subtracting this from all measurements, and (3) applying a linear fit of the type $C(E - E_{gap})$ to the measurements in the interval of −0.15 eV to −1.35 eV relative to the dip of $TiO_2$ and the interval of −0.3 eV to −1.95 eV relative to the dip in ZnO. The band gap measurements were carried out for all MOx/Si heterojunctions, i.e., for all layer thicknesses of both $TiO_2$/Si and ZnO/Si heterojunctions (see Fig. 7b).

The determination of optical bandgaps was done using the classical method, where the absorption coefficient is obtained from the measured extinction coefficient as $\alpha = 4\pi\kappa/\lambda$[60]. For predominantly crystalline semiconductors with indirect bandgap, such as anatase $TiO_2$, the bandgap is obtained from a plot of $\sqrt{\alpha}$ vs the photon energy ($h\upsilon$) as the extrapolation of the linear region to the line representing the background absorption as shown in Supplementary Fig. 6. For ZnO, which has a direct bandgap, the same procedure is applied, except that here, the linear region of $\alpha^2$ is extrapolated to the zero absorption point as no background subtraction is required. The error estimation is based on the errors propagated from the linear fits, where the fitting procedure yielding the slopes and axis intersections of the straight lines also yields the errors on slopes and axis intersections.

## Pre-treatment/annealing of samples

Annealing for pre-treatment of samples was performed in a custom build annealing jig made from two 8 mm thick aluminum plates, one comprising $25 \times 25 \times 1.5$ mm³ recesses to hold the samples, and the other was used as the lid. Both parts of the jig comprised four holes for fitting a temperature sensor. The temperature was measured with a calibrated PT100 temperature sensor (RS components) in both the sample holder and lid by 4-wire resistance measurement with an Agilent 34410 A multimeter. The jig was heated by placing it on a hotplate (Schott Instrument SLR 00997974). The final temperature of the jig reached 105(5) °C (average from all 8 measuring points on the jig). This temperature corresponded to a setting on the hotplate and was used to anneal all samples for 45–60 min prior to UV exposure and wetting

measurements. After exposure, recovery of the initial contact angle was also obtained by the same heating procedure. Alternatively, dark storage could also be used for recovery of the original wetting state. However, this took a considerably longer time, counted in days.

## UV sources for photoinduced switching

The samples were illuminated with UV LED Lamps of wavelengths 275, 300, 365, and 405 nm procured from Thorlabs Inc (Item no. M275L4, M300L4, M365L3, and M405L4 respectively). The lamps produce stable output powers of up to 80, 24, 1000, and 1000 mW, respectively, based on the maximum current recommended in the datasheets. For higher power, we procured new lamps and tested them with higher currents for their stability to achieve a reliable power output of up to 175 mW for the 275 nm DUV source. This was required for optimal photocatalytic/induced switching of ZnO/Si samples that require significantly higher doses to switch compared to $TiO_2$/Si. The DUV LED lamps of 275 and 300 nm were equipped with a high-NA collimating lens made of UV fused silica material for high transparency (ASL2520-UV from Thorlabs) and optimal power output. A regular fused silica lens of the same specifications was used for the 365 nm lamp. During illumination, the temperature was maintained constant to within 2 °C. A rise in temperature of 2 °C will, however, only decrease the surface tension of water by ~1% and has a negligible effect on the measured contact angles.

## UV dose quantification

All the lamps and collimators were assembled to achieve collimated light of approximately 29(2) mm in diameter at 6–8 mm distance from the lens. The lamps' power ($P$) was measured with a PM100D power meter and an S425C thermal power sensor head (Thorlabs Inc.). The area ($A_b$) of the light beam on the sample was calculated from the measured diameter ($d_b$) of the spot size of the illumination light. The intensity ($I_0$) is defined as the ratio of power and area. The errors were estimated as standard error accumulation, with $I_0 = P/A_b$, and $\delta I_0 = (4/\pi d_b^2)\sqrt{\delta P^2 + 4P^2(\delta d_b)^2/d_b^2}$. The illumination time was measured in seconds using a smartphone timer, and the dose $D_b$ was calculated by multiplying the intensity by the time. $D_b = I_0 \cdot t$, with standard error $\delta D_b = \sqrt{(I_0 \cdot \delta t)^2 + (t \cdot \delta I_0)^2}$. The following uncertainties were applied: $\delta t = 2$ s and $\delta d_b = 1$ mm, while $\delta P$ was obtained as the difference in the power before and after exposure was measured for 5 min and averaged.

## Lab environment

Temperature, humidity, and pressure were measured with Rotronic BL-1D (RS Components) before and after experiments, and the average and standard deviations were found from the measurements. The goniometer measurements were done in 21.7(1) °C temperatures, 1004(17) hPa pressures and an humidity of 33(9) Rh%

## Contact angle measurements

A contact angle goniometer (Attension Theta optical tensiometer, Biolin Scientific procured from BergmanLabora AB, Sweden) was used for contact angle measurements. Droplet shapes were fitted using the Young-Laplace fitting method (One Attention software)[61]. For the wet exposure experiments, the dispenser unit of the goniometer was dismounted, and the UV LED lamps were assembled on top of a goniometer stage, where the sample was placed underneath it (see Supplementary Fig. 4). A water droplet of 20 μl was placed on the sample and exposed to UV light simultaneously with the recording of the contact angle through the camera of the goniometer. With this method, the water stayed on the sample, which switched from a hydrophobic to a hydrophilic state during exposure. In the second method, the goniometer dispensing unit was used to obtain the

**Table 1 | Fitting conditions for fitting of switching time constants by Eq. 4**

|  | Minimum | Initial | Maximum limit |
|---|---|---|---|
| $\tau$ | 1 s | 80 s | Infinite |
| $\theta_0$ | Minimum $\theta_0$ in the used datasets | Average $\theta_0$ in the used datasets | Maximum $\theta_0$ in the used datasets |
| $\theta_1$ | 1° | 12° | 15° |
| $\triangle t$ | 0 | 0 | Time of the first photo-induced switch |

advancing- and receding contact angle by employing the traditional inflation/deflation method[62]. Here, the samples were pre-exposed in ambient conditions to the UV light for specific time intervals and investigated on the goniometer for the switching of their wetting state thereafter. Based on the difference in the methods, the first one is referred to as the wet exposure method, and the latter is referred to as the dry exposure method in this paper.

## Analysis of Goniometer data

A Python script was used to analyze the data obtained from the goniometer. Datapoints outside $1° < \theta < 100°$ and 1 mm < baseline < 10 mm were discarded. The remaining data points were compared to the two preceding points and the two next points. If the contact angle was 1° and/or baseline 0.1 mm larger or smaller than all datapoints compared, then the datapoint was discarded as well.

## Dry exposure contact angle analysis

Valid data points for the advancing and receding contact angle were recorded if the baseline width differed from the initial measurement by at least 300 μm. This was to ensure that the expansion/retraction was stable. Secondly, we required that the baseline must change a minimum of 20 μm from the previous data point. All valid data points for measurement were averaged, and the standard deviation was calculated with the Python NumPy package. For evaluating repeated experiments, the weighted average method was used by weighting the result by the smallest uncertainties.

## Wet exposure contact angle analysis

During wet exposure, contact angle recording $\theta_0$ was defined as the maximum contact angle in the first 30 datapoints, and any other datapoints with higher contact angle than $\theta_0$ were discarded. Datapoints were considered valid when (1) the recorded contact angle was at least 1° lower than the previous datapoint, and 0.05 mm larger in baseline than the previous 5 accepted datapoints, and when the previous three datapoints were accepted. All valid datapoints from repeated experiments were combined into a single sorted set of values and fitted with the model (Eq. 4). The fit was performed with the SciPy packages Curve_fit function, allowing up to 5 million iterations and the following bounds and initial conditions shown in Table 1. Uncertainties of the fitting parameters were obtained by taking the square root of the diagonal in the fit's covariance matrix.

## Data availability

The data that supports the findings of the study are included in the main text and supplementary information files. Source Data files (goniometer, REELS, and AFM, sorted after figure appearance in the publication) have been deposited in the repository under accession code https://doi.org/10.11583/DTU.24573367.v2[63].

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

## Acknowledgements

We would like to acknowledge the Villum Foundation for financial support to the project "LIRN Fluidics" under the "Villum Experiment Program" with grant number 40692. R.A. Deshpande acknowledges financial support from the "Villum International Postdoc Program" with grant number 51472. Finally, the authors acknowledge DTU Nanolab for access to cleanroom and characterization facilities.

## Author contributions

R.A.D.: experimental investigation (lead), methodology (contributed), data analysis (contributed), visualization (contributed), writing—original draft (contributed), writing and revision (contributed), sample fabrication (contributed). J.N.: experimental investigation (lead), methodology (contributed), data analysis (lead), visualization (contributed), writing—original draft (contributed). M.V.A.: sample fabrication (lead). E.S.: experimental investigation (contributed), sample fabrication (contributed), methodology (contributed). A.C.: data analysis (contributed), methodology (contributed), writing original draft (contributed), writing and revision (contributed). O.H.: methodology (contributed), data analysis (contributed), theory (contributed), writing, and revision (contributed). J.B.: methodology (contributed), supervision (contributed), writing, and revision (contributed). R.T.: conceptualization (lead), funding acquisition (lead), data analysis (contributed), methodology (lead), theory (lead), supervision (lead), writing—original draft (lead), writing and revision (lead).

## Competing interests

The authors declare no competing interests.
