## [Peer Review File · Nature Communications]

Understanding the light induced hydrophilicity of metal-oxide thin filmsREVIEWER COMMENTS

Reviewer #1 (Remarks to the Author):

This paper provides a physical examination of the phenomenon of photo-induced superhydrophilicity in titanium dioxide and zinc oxide. The model proposed in this paper could potentially be applicable to the photo-induced hydrophilicity of general photocatalysts. So it could be particularly valuable for the applications of photocatalysis. In order to publish this paper in Nature Communications, the referee suggests that the authors incorporate the following considerations.

1. How was the effect of silicon substrate, especially in the case of ultrathin TiO₂ and ZnO? If the thickness is thin, the effect of the excitation in Si would not be negligible, since light would reach to a substrate.
2. Did the authors consider the roughness factor of their film in their model? A practical film has roughness. The water contact angle on a rough surface is represented by Wenzel's equation.
3. The authors mentioned "most of the reported experimental studies of the aforementioned effects have been carried out with mercury lamps generating spectra comprising multiple lines in the UV range." in their introduction section. However, this sentence is not appropriate, since many studies used a fluorescent bulb and/or a black light bulb for the test of super-hydrophilicity (for example, Adv. Mater. 12, 1923, 2000).

Reviewer #2 (Remarks to the Author):

Comments on the manuscript (MS No.: NCOMMS-23-37027) titled as "Understanding the light induced hydrophilicity of metal-oxide thin films".

This manuscript reports the theoretical modeling of surface wettability of TiO₂ and ZnO under UV light irradiation. This manuscript encompasses intriguing revelations for the scholarly readership of the Nature Communications. Nevertheless, I am wondering the broad applicability of this modeling to all TiO₂ and ZnO thin films synthesized through alternative methodologies.

- 1) In my experience, the TiO₂ thin films synthesized through physical techniques in a high-vacuum chamber, such as sputtering and electron-deposition, utilizing inorganic sources, exhibit rapid manifestation of heightened wettability under UV light irradiation due to minimal surface contamination. That is, the surface remains highly clean even prior to UV light irradiation. In contrast, thin films

synthesized via the sol-gel method utilizing organic solvents do not exhibit a prompt responsiveness in surface wettability upon light irradiation. This phenomena can be readily expounded through the concentration of surface contamination by organic compounds, thereby inducing reduced wettability (evidenced by high contact angles of water droplets).

2) The modeling is represented as the concentration of “inactivated” and “activated” surface groups. As a chemist, particularly specializing in photocatalysts, I am intrigued by the nature of these surface groups. It is conceivable that the authors might not have elucidated the specific surface species, given the intricate and multifaceted nature of photocatalyst surfaces. Nevertheless, if feasible, the authors should mention the potential sources of “inactivated” and “activated” surface groups.

3) The phenomenon is known as a photo-induced hydrophilicity, signifying the increasing in hydrophilicity of the photocatalyst surface by UV light radiation. This might be rephrased that the adsorption amount of H₂O molecules increases after UV light irradiation. Is it correct? However, when the solid surface is irradiated with light, an inevitable consequence is the escalation of temperature. Why does the adsorption of H₂O molecules increase on a surface that has undergone temperature elevation? In fact, when the TiO₂ surface was subjected to UV light irradiation, the adsorption amount of H₂O molecules decreased.

This represents a straightforward phenomenon, like a drying of wet cloths in sunlight rather than in the shade. Even with stringent temperature control over the sample surface, how can we control the non-radiative recombination of the photo-formed electron-hole pairs? If so, many researchers may reply that the samples should be irradiated under cooling. In this case, the adsorption of H₂O indeed intensifies, but it is attributed to a temperature effect.

4) Under the conditions of “dry” exposure, the thin film samples underwent UV light irradiation, after which the contact angles of water droplets were measured. In contrast, in the “wet” exposure, the thin films bearing water droplets were irradiated with UV light. Is this correct? The contact angles of water droplets are generally measured by approximately 0.8-1.0 μ L of water via a micro-syringe. Throughout the “dry” exposure, the water droplet volume remains constant. However, when the thin films with water droplets were irradiated with UV light, the elevated temperature triggers the vaporization of water droplets on the surface. Moreover, even a marginal temperature elevation within the water droplets engenders a reduction in the intermolecular hydrogen bond network, thereby decreasing in the contact angles. In essence, the phenomenon, particularly concerning the adsorption of H₂O, is remarkably sensitive to fluctuations in temperature. If the theoretical modeling and experimental results in this ms can completely and successfully be explained without such temperature-driven effects, the accrued results could indeed bear noteworthy and substantial significance.

5) The photo-induced surface wettability was initially reported in the publication in “Nature, 388, 431–432 (1997)” by Fujishima et al. At least, this work should be appropriately referenced within the scope of this manuscript.

Reviewer #3 (Remarks to the Author):

This manuscript presents an elaborated and thorough study which establishes a demanding relationship between the change in contact angle at illuminated metal oxide surfaces and the charge carrier diffusion length. While the dependence of the contact angle change on several factors such as light intensity, light energy, layer thickness, and band bending has been already reported in separated studies, the correlation of the contact angle change to the surface diffusion of the photo-generated charge carriers upon control of the aforementioned factors is novel. Hence, this study has a strong impact in the field of photocatalysis and I recommend the publication after minor revision. Below are detailed comments which need to be addressed.

- The manuscript contains at several position (e.g. page 3, line 88) expression “viz” the meaning of which is not clear in the context.
- Have the authors measured the crystallographic nature of the films, which should be possible for the thicknesses larger than 50 nm. Could the authors comment how the crystallinity degree and the surface to bulk ratio change with increase of the thickness layer and which impact this might have on the contact angle results.
- In Figure 2 the hole concentration is given in arbitrary units, could the authors explain it the text.
- Page 4, line 117-119: Due to the n-type doping the authors assumed an excess in electron concentration. Could the authors estimate the ratio of the “dark” electrons to the photo-generated electrons and thus control if the assumption is valid for all used illumination doses.
- Page 9, 284-288: By the discussion of the wet vs. dry condition it needs to be considered that also under dry conditions multilayers

REVIEWER COMMENTS

In the following, we address point-by-point the revision requests and comments by all three reviewers. The corresponding changes in the manuscript and the Supplementary Materials are marked in yellow.

Reviewer #1 (Remarks to the Author):

This paper provides a physical examination of the phenomenon of photo-induced superhydrophilicity in titanium dioxide and zinc oxide. The model proposed in this paper could potentially be applicable to the photo-induced hydrophilicity of general photocatalysts. So it could be particularly valuable for the applications of photocatalysis. In order to publish this paper in Nature Communications, the referee suggests that the authors incorporate the following considerations.

1. How was the effect of silicon substrate, especially in the case of ultrathin TiO₂ and ZnO? If the thickness is thin, the effect of the excitation in Si would not be negligible, since light would reach to a substrate.

Answer: Part of the optical power is, indeed, absorbed in the silicon, but we have compensated for that by calculating the absorbance in MO_x only, and used that in the calculations.

However, photoexcitation of e-h pairs in the Si may play a role for thin MO_x layers. Due to the staggered band alignment, photoexcited electrons in Si may transfer to the MO_x, whereas the potential barrier at the interface will reflect most of the holes. Some holes from the MO_x may recombine with electrons transferred from the MO_x. The electrons transferred from the Si side to the MO_x will likely recombine with the excess holes generated in the MO_x, and contribute to the assumed boundary condition $p(w) = 0$. Therefore, this effect does not cause any deviation from our model. However, as a secondary consequence it might change the space charge layer thickness w . For simplicity, our model does not take this minor aspect into account.

2. Did the authors consider the roughness factor of their film in their model? A practical film has roughness. The water contact angle on a rough surface is represented by Wenzel's equation.

Answer: Yes, this is a fair comment, and the surface roughness was measured by AFM for the fabricated films (Supplementary figure S1), and was generally of order 5 nm. The effective surface area was also obtained by AFM yielding roughness factors (real surface area to projected surface area) in the Wenzel model of $r \sim 1.03$ for TiO₂ and $r \sim 1.05$ for ZnO. Only for contact angles below $\sim 30^\circ$ errors larger than the sample to sample variations caused by the dominating pinning effects (contact angle hysteresis) would be noticeable if the Wenzel model was considered. Hence, this extra complication would only have an effect for few data points. For simplicity, we decided to disregard the effect of surface roughness, although for rougher surfaces it would be possible to include this in the model. Including it would require that the Wenzel roughness factor is determined for each surface prior to the contact angle measurements.

3. The authors mentioned "most of the reported experimental studies of the aforementioned effects

have been carried out with mercury lamps generating spectra comprising multiple lines in the UV range." in their introduction section. However, this sentence is not appropriate, since many studies used a fluorescent bulb and/or a black light bulb for the test of super-hydrophilicity (for example, Adv. Mater. 12, 1923, 2000).

Answer: Indeed UV illumination was carried out using a 10 W fluorescent light bulb in the quoted paper by Miyauchi et al.. Although a fluorescent light bulb is different from a mercury lamp, it still does not feature a narrow Gaussian single-line spectrum. However, we have corrected the wording and cited the proposed reference.

Reviewer #2 (Remarks to the Author):

Comments on the manuscript (MS No.: NCOMMS-23-37027) titled as "Understanding the light induced hydrophilicity of metal-oxide thin films".

This manuscript reports the theoretical modeling of surface wettability of TiO₂ and ZnO under UV light irradiation. This manuscript encompasses intriguing revelations for the scholarly readership of the Nature Communications. Nevertheless, I am wondering the broad applicability of this modeling to all TiO₂ and ZnO thin films synthesized through alternative methodologies.

1) In my experience, the TiO₂ thin films synthesized through physical techniques in a high-vacuum chamber, such as sputtering and electron-deposition, utilizing inorganic sources, exhibit rapid manifestation of heightened wettability under UV light irradiation due to minimal surface contamination. That is, the surface remains highly clean even prior to UV light irradiation. In contrast, thin films synthesized via the sol-gel method utilizing organic solvents do not exhibit a prompt responsiveness in surface wettability upon light irradiation. This phenomena can be readily expounded through the concentration of surface contamination by organic compounds, thereby inducing reduced wettability (evidenced by high contact angles of water droplets).

Answer: We thank the reviewer for the insightful comments. We believe that the phenomenon of surface contamination was also encountered in this work, as a time delay was observed (see figure 4b) before onset of switching. In this paper, we attribute this delay to the time it takes to remove hydrocarbons from the surface. No changes are requested from the reviewer in this point.

2) The modeling is represented as the concentration of "inactivated" and "activated" surface groups. As a chemist, particularly specializing in photocatalysts, I am intrigued by the nature of these surface groups. It is conceivable that the authors might not have elucidated the specific surface species, given the intricate and multifaceted nature of photocatalyst surfaces. Nevertheless, if feasible, the authors should mention the potential sources of "inactivated" and "activated" surface groups.

Answer: We have added a sentence in the introduction. “The inactive groups are the Ti—O—Ti bridges. When a water molecule chemisorbs across one Ti—O bond, then one obtains two terminal Ti—OH groups, and those are what we call active surface groups”. This information is readily available in the literature.

3) The phenomenon is known as a photo-induced hydrophilicity, signifying the increasing in hydrophilicity of the photocatalyst surface by UV light radiation. This might be rephrased that the adsorption amount of H₂O molecules increases after UV light irradiation. Is it correct? However, when the solid surface is irradiated with light, an inevitable consequence is the escalation of temperature. Why does the adsorption of H₂O molecules increase on a surface that has undergone temperature elevation? In fact, when the TiO₂ surface was subjected to UV light irradiation, the adsorption amount of H₂O molecules decreased.

This represents a straightforward phenomenon, like a drying of wet cloths in sunlight rather than in the shade. Even with stringent temperature control over the sample surface, how can we control the non-radiative recombination of the photo-formed electron-hole pairs? If so, many researchers may reply that the samples should be irradiated under cooling. In this case, the adsorption of H₂O indeed intensifies, but it is attributed to a temperature effect.

Answer: Non-radiative recombination is included in the model by including τ_r , the excess carrier recombination lifetime. The switching rate (Eq. 7) is expressed as a function of this lifetime, so non-radiative recombination in the MO does not have to be absent for the model to apply.

Indeed the samples absorb the optical power from the LEDs. However, we have checked this and monitored the sample heating under UV illumination by an infrared camera. Here, we found that the temperature maintained constant to within 2 degrees during the experiments. A rise in temperature of 2 degrees will, however, only decreases the surface tension of water by ~1%, which in no way can give rise to the observed decrease in contact angle under illumination. We have added a description of this in the methods section.

4) Under the conditions of “dry” exposure, the thin film samples underwent UV light irradiation, after which the contact angles of water droplets were measured. In contrast, in the “wet” exposure, the thin films bearing water droplets were irradiated with UV light. Is this correct? The contact angles of water droplets are generally measured by approximately 0.8-1.0 μL of water via a micro-syringe. Throughout the “dry” exposure, the water droplet volume remains constant. However, when the thin films with water droplets were irradiated with UV light, the elevated temperature triggers the vaporization of water droplets on the surface. Moreover, even a marginal temperature elevation within the water droplets engenders a reduction in the intermolecular hydrogen bond network, thereby decreasing in the contact angles. In essence, the phenomenon, particularly concerning the adsorption of H₂O, is remarkably sensitive to fluctuations in temperature. If the theoretical modeling and experimental results in this ms can completely and successfully be explained without such temperature-driven effects, the accrued results could indeed bear noteworthy and substantial significance.

Answer: Again, we believe this comment is related to a concern about potential heating effects from the illumination and our answer is given above. In addition, as mentioned in the methods section, droplets of 20 μL and not 0.8-1.0 μL of water were used for the wet exposure experiments.

5) The photo-induced surface wettability was initially reported in the publication in “Nature, 388, 431–432 (1997)” by Fujishima et al. At least, this work should be appropriately referenced within the scope of this manuscript.

Answer: We thank the reviewer for drawing our attention to this important paper, which is now cited in the revised manuscript.

Reviewer #3 (Remarks to the Author):

This manuscript presents an elaborated and thorough study which establishes a demanding relationship between the change in contact angle at illuminated metal oxide surfaces and the charge carrier diffusion length. While the dependence of the contact angle change on several factors such as light intensity, light energy, layer thickness, and band bending has been already reported in separated studies, the correlation of the contact angle change to the surface diffusion of the photo-generated charge carriers upon control of the aforementioned factors is novel. Hence, this study has a strong impact in the field of photocatalysis and I recommend the publication after minor revision. Below are detailed comments which need to be addressed.

-The manuscript contains at several position (e.g. page 3, line 88) expression “viz” the meaning of which is not clear in the context.

Answer: We have replaced the latin abbreviation “viz.” with the more commonly used English word “namely”.

- Have the authors measured the crystallographic nature of the films, which should be possible for the thicknesses larger than 50 nm. Could the authors comment how the crystallinity degree and the surface to bulk ratio change with increase of the thickness layer and which impact this might have on the contact angle results.

Answer: To address this comment, we have conducted grazing-incidence XRD measurements on a relevant selection of the films. The results are shown in Figure S2 (Supplementary Material) and are also discussed in the main article. First of all, these XRD measurements show that TiO₂ films are in the anatase structure and ZnO films are in the wurtzite structure, as expected. Second, XRD measurements indicate that there are no significant differences in crystalline quality between the thinner and the thicker films. We draw this conclusion because there are no significant differences in the FWHM of the peaks at low and high thickness (similar crystallite size), and the peak intensity roughly doubles as the film thickness doubles (similar fraction of crystallized material within the film). Thus, differences in contact angle and photocatalytic activity with film thickness cannot be ascribed to differences in crystallinity with thickness.

For the impact of surface to bulk ratio changes on contact angle results, please see Figure S1, Supplementary Material, and the answer to Comment #2 by Reviewer #1.

- In Figure 2 the hole concentration is given in arbitrary units, could the authors explain it the text.

Answer: We have replaced figure 2 with one in which the hole concentration is plotted in units of $(I_0 A)/(h\nu D w)$, i.e. what is plotted in the figure is the dimensionless function $L^2/wd (1 - 1/\cosh((d - w)/L))$, with $w = 20$ nm. Hence, now the hole concentration at the surface is plotted in units that are not varied in the plots, namely the diffusion length L and MOx thickness d . We have added a suitable description in the figure caption.

- Page 4, line 117-119: Due to the n-type doping the authors assumed an excess in electron concentration. Could the authors estimate the ratio of the “dark” electrons to the photo-generated electrons and thus control if the assumption is valid for all used illumination doses.

Answer: Since electrons and holes are photogenerated in pairs, the photoinduced electron concentration will be of order, $\sim(I_0 \tau_r)/h\nu d$. The issue with this estimation is that the recombination time is not known, but for $I_0 \sim 100$ mW/cm², $d \sim 50$ nm, a photon energy $h\nu \sim 3$ eV, and assuming a relatively long recombination time of order 1 μ s, we get $p \sim 5 \times 10^{16}$ cm⁻³, which is two orders of magnitude lower than the native doping level in TiO₂. Given that the recombination is in fact likely faster than 1 μ s, p is likely to be much lower and the assumption introduces no significant error. We have added a remark about this in the revised manuscript.

- Page 9, 284-288: By the discussion of the wet vs. dry condition it needs to be considered that also under dry conditions multilayers

Answer: We are not sure what is asked for here. It seems like the last comment by reviewer 3 got truncated during a copy-paste action. We guess this has to do with a proposal to provide a full analysis of the switching rate for the dry exposure set. However, a full analysis is not feasible due to very weak switching at dry exposures, especially for ZnO samples that would require a multitude of high doses, i.e. illuminations for very long times to acquire the individual data points. The beauty of the wet exposure method is that contact angle datapoints are recorded during the exposure, while the dry exposure method requires annealing for one hour between each contact angle measurement. For a thorough comparison between dry and wet exposure contact angle switching, a completely new batch of heterostructure samples with different MOx layer thicknesses on substrates featuring faster switching would be required. Such a study is left for a follow-up paper.

REVIEWER COMMENTS

Reviewer #1 (Remarks to the Author):

The paper has properly been revised based on the reviewers' comments. It is publishable as is.

Reviewer #2 (Remarks to the Author):

Comments on the manuscript (MS No.: NCOMMS-23-37027A) titled as "Understanding the light induced hydrophilicity of metal-oxide thin films".

This manuscript is now well revised according to the reviewer's comments. So I think it can be accepted for "Nature Communications", just after one point will become clear.

1) In the preceding review, I sought elucidation regarding the origin of "inactivated" and "activated" surface groups. The authors, in their response, explicated that these groups correspond to the Ti-O-Ti bridges and the two terminal Ti-OH groups formed through the chemisorption of H₂O molecules onto the Ti-O-Ti bridge site. Nevertheless, in my own empirical investigations, I have yet to discern an augmentation in the hydroxyl groups on the TiO₂ surface by UV light irradiation, as determined by FT-IR measurements. Furthermore, my findings demonstrate that the adsorption of H₂O molecules onto the Ti³⁺ sites, on which photo-formed electrons are trapped, did not increase.

J. M. White et al., also mentioned UV irradiation of a hydrophobic surface, which is covered with TMA (trimethyl acetate), in the presence of O₂ removes TMA and rapidly restores hydrophilicity. They also asserted that the presence of oxygen atom vacancies did not appreciably perturb the hydrophilicity of either pristine or TMA-covered TiO₂(110) surfaces, as elucidated in their work (J. Phys. Chem. B 2003, 93, 34, 9029–9033).

If the augmentation of surface hydroxyl groups is really responsible for photo-induced surface wettability, it is imperative to scrutinize experimental evidence substantiating this claim, such as via FT-IR measurements. I acknowledged much of the literature that delves into the mechanisms underpinning this phenomenon. However, the authors have rebutted the influence of temperature variations induced by light irradiation because the temperature changes within 2 °C result in a marginal reduction of water surface tension of less than 1%. Consequently, this insight prompts the question of whether a relatively minor augmentation in surface hydroxyl groups induced by UV light, as reported in certain literature, can sufficiently account for macroscopic shifts in surface wettability. If the authors claim that the hydroxyl groups increase by light irradiation on TiO₂ surface, they should not only cite literatures but also show experimental data to support their assertion.

Reviewer #3 (Remarks to the Author):

The authors have addressed the comments made by the reviewers in the revised version of the manuscript. Therefore, I have no further comments and the manuscript can be published as it is.

REVIEWER COMMENTS

Reviewer #1 (Remarks to the Author):

The paper has properly been revised based on the reviewers' comments. It is publishable as is.

Answer: We appreciate the reviewer's statement.

Reviewer #2 (Remarks to the Author):

Comments on the manuscript (MS No.: NCOMMS-23-37027A) titled as "Understanding the light induced hydrophilicity of metal-oxide thin films".

This manuscript is now well revised according to the reviewer's comments. So I think it can be accepted for "Nature Communications", just after one point will become clear.

1) In the preceding review, I sought elucidation regarding the origin of "inactivated" and "activated" surface groups. The authors, in their response, explicated that these groups correspond to the Ti-O-Ti bridges and the two terminal Ti-OH groups formed through the chemisorption of H₂O molecules onto the Ti-O-Ti bridge site. Nevertheless, in my own empirical investigations, I have yet to discern an augmentation in the hydroxyl groups on the TiO₂ surface by UV light irradiation, as determined by FT-IR measurements. Furthermore, my findings demonstrate that the adsorption of H₂O molecules onto the Ti³⁺ sites, on which photo-formed electrons are trapped, did not increase.

J. M. White et al., also mentioned UV irradiation of a hydrophobic surface, which is covered with TMA (trimethyl acetate), in the presence of O₂ removes TMA and rapidly restores hydrophilicity. They also asserted that the presence of oxygen atom vacancies did not appreciably perturb the hydrophilicity of either pristine or TMA-covered TiO₂(110) surfaces, as elucidated in their work (J. Phys. Chem. B 2003, 93, 34, 9029–9033).

If the augmentation of surface hydroxyl groups is really responsible for photo-induced surface wettability, it is imperative to scrutinize experimental evidence substantiating this claim, such as via FT-IR measurements. I acknowledged much of the literature that delves into the mechanisms underpinning this phenomenon. However, the authors have rebutted the influence of temperature variations induced

by light irradiation because the temperature changes within 2 °C result in a marginal reduction of water surface tension of less than 1%. Consequently, this insight prompts the question of whether a relatively minor augmentation in surface hydroxyl groups induced by UV light, as reported in certain literature, can sufficiently account for macroscopic shifts in surface wettability. If the authors claim that the hydroxyl groups increase by light irradiation on TiO₂ surface, they should not only cite literatures but also show experimental data to support their assertion.

Answer: We appreciate the reviewer's statement regarding our last revision. Regarding the remaining point, we respect the reviewer's position that the relevant chemical reaction is debatable, and our intention was only to convey a possible unitary reaction found in the literature. We hope this position is made clearer now.

Reviewer #3 (Remarks to the Author):

The authors have addressed the comments made by the reviewers in the revised version of the manuscript. Therefore, I have no further comments and the manuscript can be published as it is.

Answer: We appreciate the reviewer's statement.